# Verifiable Federated Learning

**Simone Bottoni**
University of Insubria
Varese, Italy
sbottoni@uninsubria.it

**Giulio Zizzo**
IBM Research Europe
Dublin, Ireland
giulio.zizzo2@ibm.com

**Stefano Braghin**
IBM Research Europe
Dublin, Ireland
stefanob@ie.ibm.com

**Alberto Trombetta**
University of Insubria
Varese, Italy
alberto.trombetta@uninsubria.it

## Abstract

In Federated Learning (FL) a significant body of research has focused on defending against malicious clients. However, clients are not the only party that can behave maliciously. The aggregator itself may tamper the model to bias it towards certain outputs, or adapt the weights to aid in reconstructing a client's private data. In this work we tackle the open problem of efficient verification of the computations performed by the aggregator in FL. We develop a novel protocol which through using binding commitments prevents an aggregator from modifying the resulting model, and only permits the aggregator to sum the supplied weights. We provide proof of correctness for our protocol demonstrating that any tampering by an aggregator will be detected. Additionally, we evaluate our protocol's overheads on three datasets, and show that even for large neural networks with millions of parameters the commitments can be computed in under 20 seconds.

## 1 Introduction

Federated Learning (FL) is a new paradigm for training machine learning (ML) models in a decentralised fashion. Clients, who collect and hold data, will locally train ML models. Once local training has been completed, the updated models are sent to the aggregator to be combined. There has been a large amount of research examining how to protect FL systems against malicious clients, for example, by using robust aggregation schemes [1], examining client behaviour over a window of time [2], or leveraging certification algorithms [3, 4].

However, clients are not the only party that could misbehave. The aggregator itself may tamper with the aggregated model. For example, in product recommendation a service provider wishes to optimise for maximum profits, while a client wishes to see items that fulfils their criteria while minimising expense. Therefore, clients may wish to verify that a service provider is not biasing the aggregated model towards more expensive products. Alternatively, a malicious aggregator can adapt the weights to create so called "trap weights" to launch data reconstruction attacks against a client [5]. Hence, a client's ability to verify that an aggregator performed the operations it claimed to conduct is of significant security importance.

This threat model of a malicious aggregator has a significantly more limited body of research compared to the traditional malicious client threat model. Papers have proposed homomorphic hashes [6] or setups based on Pallier encryption [7, 8]. In this work, we propose a lightweight verification protocol we call Verification Via Commitments (VVC) that enables clients to verify

Workshop on Federated Learning: Recent Advances and New Challenges, in Conjunction with NeurIPS 2022 (FL-NeurIPS'22). This workshop does not have official proceedings and this paper is non-archival.

the aggregator, even when it colludes with malicious clients. More precisely, our contribution is a lightweight protocol that:

- Enables clients to verify that the aggregator correctly performed sums over the supplied weights and that the weights have not been tampered with.

- Is resistant to a configurable number of colluding clients with a malicious aggregator via Shamir's Secret Sharing.

- Maintains high Federated Learning accuracy and is scalable to large neural networks, typically handling millions of parameters in under 20 seconds.

## 2 Related Works

Reducing the trust placed in cloud services for ML has a growing body of work as ML models are being deployed and trained in an ever more distributed fashion. An early work in this area is SafetyNets [9] which used an Interactive Proof protocol [10] to verify that a neural network's forward pass was correctly executed by a service provider. Converting this style of verification for FL aggregation requires the aggregator and clients needing to communicate many times for the verification protocol. Hence, alternative strategies more suited to FL have since been developed [11].

One strand of research has used the idea of homomorphic hash functions [6, 12], which can provide verification that an aggregator has correctly performed a sum of supplied gradients, albeit with non-trivial overhead in verification time. Alternative strategies have been proposed in [13] with clients encrypting gradients using a pseudorandom generator and Lagrange interpolation, together with the Chinese Remainder Theorem. The aggregator only sees, and operates on, the encrypted gradients and the clients perform verification by unpacking the aggregated ciphertext they receive.

The above works make the assumption that malicious clients can try and collude with the aggregator to break the verification schemes. Hence, they often use a trusted third party to initialise the initial key generation. This requirement has been relaxed other works such as [7, 14] but often carries some penalty: for example in [7] the clients are assumed to be honest and not collude with the aggregator. Recent works have started looking at ways to reduce the trust in a key-generator, for example [12] assumes a key-generator that is truthful but inquisitive.

## 3 Verification Via Commitments (VVC) protocol

We present Verification Via Commitments (VVC), a novel and lightweight verification protocol that can be applied to FL. Our protocol allows clients to continuously, or periodically, verify that the aggregator performs the aggregation correctly over the supplied weights and that the weights were not tampered with. Our protocol also enables clients to verify the aggregator computation even when it colludes with some clients, up to a maximum of all clients minus one.

In the following sections, we describe how our protocol works, and the Cryptographic primitives used. We present the components that are part of the system and all the operations that must be performed to train a FL model and verify that it has been calculated correctly.

### 3.1 Cryptographic Primitives

#### 3.1.1 Pedersen Commitments

Our protocol is based on the Pedersen commitment scheme [15]. A commitment scheme is a cryptographic primitive that, using a secret random value, allows one to commit a given value. Only parties knowing both values can open the commitment.

The Pedersen Commitment scheme is a homomorphic commitment based on the discrete logarithm problem over a cyclic group $G$. We base our approach on modular arithmetic and consider the case where $G$ is a sub-group $G$ of order $q$ of $\mathbb{Z}_p$, where $p, q$ are both (large) primes and $q$ divides $p - 1$. The Pedersen Commitment scheme uses two random generators of $G$, $g$ and $h$, to compute the commitment of a value.

A committer chooses a value $w$, a random value $r$, and computes the commitment:

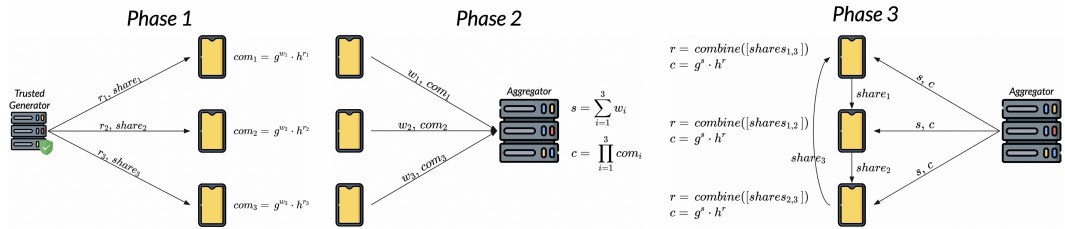

Figure 1: Protocol Phases.

$$c = g^w * h^r \tag{1}$$

The commitment $c$ can be opened only by revealing both $w$ and $r$. No one can open the commitment without knowing both $w$ and $r$ unless it can compute $log_g(h)$ which requires solving the discrete log problem over $G$. The aggregator is prevented from changing $w$ to a malicious value $w'$ unless it can solve $r' = log_h(c * g^{-w'})$ which is computationally infeasible. We say that the Pedersen Commitment scheme is perfectly hiding and computationally binding.

### 3.1.2 Shamir Secret Sharing

Our protocol uses Shamir's Secret Sharing (SSS) [16], also known as the $(k, n)$-threshold scheme, to share a secret value between clients. The purpose of SSS is to partition the data $D$ that must be maintained secret into $n$ pieces, called shares. The knowledge of $k$ or more shares makes $D$ computable, but the knowledge of any $k - 1$ or fewer pieces leaves $D$ undetermined. SSS is based on polynomial interpolation in a 2-dimensional plane. It consists of the computation of a polynomial $q(x)$ of degree $k - 1$ where the constant term $q(0)$ is $D$, and other terms are random. Given any subset of $k$ shares is possible to find the polynomial by interpolation and compute the secret $D$. On the other hand, knowledge of $k - 1$ values is not sufficient to compute the secret.

SSS is based on modular arithmetic using integers modulo $p$ as polynomial coefficients, where $p$ is a prime number. An adversary that obtains $k - 1$ shares can compute, for each possible secret value $D'$ in the interval $[0, p)$ one and only one polynomial $q'(x)$ of degree $k - 1$ such that $q'(0) = D'$. These $p$ polynomials are equally likely; the attacker obtains nothing about the real value of the secret $D$.

### 3.2 Protocol

We assume the presence of a trusted generator that computes the necessary protocol materials (public parameters and random values) and distributes them to the appropriate parties participating in the protocol execution.

The trusted generator produces the public parameters $g$, $h$, and $q$ that clients need to compute and verify a commitment, as described in Algorithm 1. In each round, it generates a random value for each client. It computes the secret for the round by summing all the generated random values and splits it into multiple shares (based on the number of clients) using the SSS protocol. The trusted generator, for each round, sends each client a random value necessary to compute commitments, the shares required to recompute the round's secret to verify the aggregator commitment, and the number of clients that participates in the round.

If the number of clients per round is pre-known, the trusted generator can pre-generate random values and pre-compute shares in the protocol setup phase and distribute them before the protocol starts.

We present an example in Figure 1. We describe the protocol in a single round since the behavior of the system's involved parties is the same for all rounds.

Our protocol consists of three different phases. In the first phase, each client trains the FL model on its local data and computes a commitment for each parameter in the model using the random value it received from the trusted environment. The computation of commitments is described in Algorithm 1. Then, all the clients send the aggregator their FL parameters and the computed commitments. In the second phase, the aggregator sums all the weights, multiplies all the commitments, and sends back the results to clients. In the third and last phase, clients verify that the aggregator correctly

**Function** `public_parameters_generation()`:
  Choose $p$, $q$ random primes such that $q$ divides $p-1$, $G_q$ is the unique subgroup of $Z_p^*$ of
    order $q$, $g$ and $h$ are generators of $G_q$.
  Select $g, h \in G_q$.
  Choose number of participating clients. $n$
  **return** $g, h, q, n$

**Function** `parameters_generation(`$n$`)`:
  The Trusted Entity selects a random number $r_i$ for each client, and it computes the sum

$$r = \sum_{i=1}^{n} r_i. \tag{2}$$

  Then it computes $n$ shares using the Shamir($k$, secret) threshold scheme [17], where $n$ is the
  number of clients and $k$ is the necessary number of shares to recompute the secret $r$.
  **return** $r_i$, $share_i$

**Function** `commit(`$g, h, q, w_i, r_i$`)`:
  Each client computes:
$$com_i = (g^{w_i} * h^{r_i}) \bmod q \tag{3}$$

  **return** $com_i$

**Function** `aggregate(`$[w_i], [com_i]$`)`:
  The Aggregator computes:

$$s = \sum_{i=1}^{n} w_i \ \text{ and } \ c = \prod_{i=1}^{n} com_i \tag{4}$$

  **return** $s, c$

**Function** `compute_secret(`$[share_i]$`)`:
  Each client computes the secret ($secret = r$) using the Shamir($k$, secret) threshold
    scheme [17]. Each client uses his own share and $k-1$ shares obtained from other clients.
  **return** $secret$

**Function** `verify(`$secret, c, s\ n$`)`:
  Each client computes:
$$com = (g^s * h^{secret}) \bmod q \tag{5}$$
  If $com = c$ accept the computation and perform $w = s/n$ , otherwise reject.
  **return**

**Algorithm 1:** Verification Via Commitments (VVC) Protocol

computed the global FL parameters. Firstly, clients re-compute the secret using SSS, exchanging at least $k$ shares between them. Secondly, a client computes the commitment of the weight calculated by the aggregator using the re-computed secret. Then it compares the computed commitment with the product of commitments received by the aggregator. Each client is convinced that the aggregator did the operation correctly if the two commitments are equal. Finally, a client will divide the weights received from the aggregator by the number of clients participating in the round.

Clients do not need to verify the aggregator computation each round. They can verify it later or periodically, storing the round's secret and the values computed by the aggregator.

### 3.3 Security considerations

**Malicious Aggregator**    We present proof that our protocol enables clients to discover a malicious aggregator that cheats or tampers the weights. We consider a scenario where an untrusted aggregator

tampers the weights, thus performing the aggregation incorrectly. We focus on Phases 2 and 3 of our protocol, where the aggregator can operate.

The malicious aggregator receives the clients' parameters and commitments, sums all the weights, and multiplies all the commitments. The malicious aggregator has two possibilities to cheat. The first is to replace one or more clients' parameters to make the training of the global FL model fail or tamper with the model in such a way to fulfill an attack objective. The second way is to replace one or more clients' parameters and re-compute the product of commitments $c$.

In the first scenario, the aggregator sends back to clients the product of commitments $c$ and a new value for the sum of parameters $s'$ such that it differs from the correct sum $s$, i.e. $s' \neq s$. Then each client re-computes the secret $r$ and perform the commitment $com = g^{s'} * h^r$. Taking into account that $s' \neq s$, the commitment will be $com \neq c$, allowing a client to discover the wrong computation.

In the second scenario we assume that the aggregator can compute a new commitment $c' = g^{s'} * h^{r'}$, such that $c' = c$ is valid. In our protocol $c = \prod_{i=1}^{n} com_i = \prod_{i=1}^{n} g^{w_i} * h^{r_i} = g^{w_1} * h^{r_1} * \ldots * g^{w_n} * h^{r_n} = g^{(w_1 + \ldots + w_n)} * h^{(r_1 + \ldots + r_n)} = g^s * h^r$ that is equal to a Pedersen Commitment. Since we took into account an aggregator able to generate a $c' = c$, so, it can generate two Pedersen Commitments $c' = g^{s'} * h^{r'}$ and $c = g^s * h^r$ such that $c' = c$ with $s' \neq s$ and $r' \neq r$.

However, if we consider [15], where the authors prove that is not possible to obtain a Pedersen Commitment $c' = c$ with $s' \neq s$ and $r' \neq r$, by extension in our protocol there does not exist an aggregator which is able to generate a new $c' = c$. Other byzantine behavior can be proven from the aggregator, for example it can try to open the commitments knowing $s$ to create a new commitment such that $c' = g^s * h^{r'}$ and $c = g^s * h^r$. Formal proofs are omitted for space constraints.

**Malicious Aggregator with colluding clients**   We present proof that our protocol is resistant to a certain number of colluding clients with a malicious aggregator. We consider a scenario where malicious clients cooperate with the aggregator to aid it in forging a tampered global FL model. Multiple clients cooperate to re-compute the secret before the aggregation and send it to the aggregator who can quickly invalidate the protocol by replacing $s$ with a $s'$ and computing a new $c = g^{s'} * h^r$.

In our protocol we introduced the SSS [17] protocol with the configurable parameter $k$, as presented in Sections 3.1 and 3.2. Given the existence of $k$, the aggregator must cooperate with at least $k$ clients to obtain the secret before the aggregation to invalidate the model successfully. In SSS $k$ is a configurable parameter, so it is possible to set it large enough (up to $k = n - 1$) to avoid the clients' collaboration. So, for FL cases subjected to a low number of malicious clients a smaller $k$ can be selected, improving the computational performance. Instead, scenarios with many malicious clients should set a large $k$ at the cost of increased computational overhead. We evaluated the correlation between the value of $k$ and the system's performance in Appendix B.

If the aggregator cooperates with $k - 1$ clients, it can obtain at most $k - 1$ shares. With the $k - 1$ shares, or any possible combination of them, it can construct one and only one polynomial $q'(x)$ of degree $k - 1$ such that $q'(0) = secret'$ with $secret \neq secret'$. As presented in 3.1.2, the polynomial terms are computed modulo $p$, so there are $p$ possible polynomials that are equally likely so that the adversary can learn nothing about the value of the secret.

# 4   Experiments

## 4.1   Federated Learning Accuracy Evaluation

Due to the common requirement in cryptography for all values to be integers, the neural network weights require rounding to a lower precision in order for commits to be computed. Concretely, this means that a copy of the weights is encoded in an integer format by fixing the precision to a given level, and multiplying by $10^b$ where $b$ is the appropriate power to give an integer result. As we can only verify to a fixed level of precision the weights which are sent to the aggregator are correspondingly rounded to the appropriate level. We empirically analysed the effect of this on three different datasets MNIST, CIFAR, and Merged-FEMNIST from the LEAF benchmark [18, 19]. The Merged-FEMNIST is the federated version of the original extended MNIST dataset [19], while merging classes corresponding to upper and lower case letters with semantically identical inputs. Following the original protocol in [18] we merged classes C, I, J, K, L, M, O, P, S, U, V, W, X, Y and

| Dataset | Regular Accuracy | Protocol Accuracy | | | |
|---|---|---|---|---|---|
| | | 2 DP | 4 DP | 6 DP | 8 DP |
| MNIST | $97.50 \pm 0.06$ | $91.67 \pm 1.16$ | $97.45 \pm 0.04$ | $97.46 \pm 0.08$ | $97.47 \pm 0.04$ |
| CIFAR | $79.72 \pm 0.16$ | $9.97 \pm 0.50$ | $79.90 \pm 0.26$ | $79.25 \pm 0.24$ | $79.36 \pm 0.41$ |
| Merged FEMNIST | $88.08 \pm 0.20$ | $53.70 \pm 4.27$ | $88.11 \pm 0.31$ | $87.79 \pm 0.66$ | $87.93 \pm 0.44$ |

Table 1: Accuracy comparison for different levels of rounding. Results are averaged over 3 runs with associated standard deviation. Higher precision results in more time required for computing commitments.

Z with their lower case versions resulting in a 47 class dataset. Full experimental description (model architecture, learning rates, etc) are found in Appendix A.

We analysed the accuracy difference over precision values of 2, 4, 6, and 8 decimal places (DP) on the three datasets considered. Over this range of precision of 4 decimal places or higher was sufficient to retain the original accuracy: differences between the original and rounded neural networks are all within a single standard deviation of each other, as show in Table 1

### 4.1.1 Time Evaluation

Performing cryptographic operations over large neural networks can incur computational overhead. We perform a sweep of the time it takes to compute commitments over all the weights in neural networks of various sizes. The results are in Figure 2.

It is only with the very largest key sizes at high levels of precision that the time requirement could be of concern. It is not unususal for cryptographic verification systems to require longer overhead, e.g [6] requires close to 100 seconds with just 5000 parameters. By comparison with our method can scale to *millions* of parameters using less time: with a key size of 1024 and avoiding excess precision the time is kept to below 20 seconds for 2.5 million weights. The security of the commitments needs to only be kept to the expected time of a FL round and so using key sizes in the range of 512-1024 will provide the required security. After a FL round has ended the clients will have the commits from the aggregator and so breaking the security of the commits at that point will not enable the aggregator to tamper with the protocol.

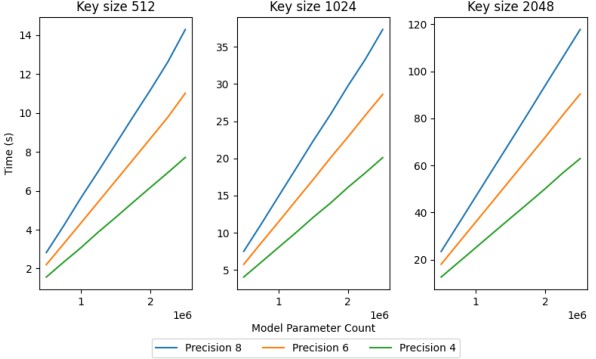

Figure 2: Time required for computing commits over model sizes ranging over 0.5 - 2.5 million parameters using different combinations of key sizes and precision. Results were collected on a Intel i7-8700 CPU.

## 5 Conclusion

In this paper we have presented Verification Via Commitments (VVC) a new protocol based on the Pedersen Commitment scheme to verify that an aggregator has correctly combined the provided weights and has not tampered with the resulting neural network. Or protocol does not interfere with the neural network performance, can be simply implemented, and is computationally lightweight by cryptographic standards.

## Acknowledgments

This work has been partially supported by the MORE project (grant agreement 957345), funded by the EU Horizon 2020 program.

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

## Appendix A   FL Setup

Here we detail the experimental configuration of the FL setups we used for Section 4. We use the following notation to describe our models: $\text{Conv}_{(s_w,s_h)}$ $C$ x $W$ x $H$ for a convolutional layer with $C$ output channels, and a kernel of width $W$ and height $H$. The stride size in width and height is given by $s_w$ and $s_h$ respectively. Likewise $\text{MaxPool}_{(s_w,s_h)}$ $W$ x $H$ for a maxpool layer with a kernel of width $W$ and height $H$. The stride size in width and height is given by $s_w$ and $s_h$ respectively. Dropout($P$) indicates a dropout layer with dropout probability $P$.

Additionally, FC $N$ indicates a fully connected layer with $N$ outputs.

### A.1   MNIST

For the MNIST dataset we used the following neural network:

$\text{Conv}_{(1,1)}$ 32x5x5 $\rightarrow$ ReLU $\rightarrow$ $\text{MaxPool}_{(2,2)}$ 2x2 $\rightarrow$ $\text{Conv}_{(2,2)}$ 64x5x5 $\rightarrow$ ReLU $\rightarrow$ $\text{MaxPool}_{(2,2)}$ 2x2 $\rightarrow$ FC 512 $\rightarrow$ ReLU $\rightarrow$ FC 10

In the MNIST dataset was randomly split between 20 clients, and each round 10 clients participated for a total of 50 FL training rounds. We use 20% of the original training data as the validation set. We used SGD as the optimiser with a learning rate of 0.01 and batch size of 32. Each FL round a participating client locally trains for one epoch before dispatching their updated weights.

### A.2   CIFAR

For the CIFAR dataset we used a VGG style network with the following architecture:

$\text{Conv}_{(1,1)}$ 32x3x3 $\rightarrow$ ReLU $\rightarrow$ $\text{Conv}_{(1,1)}$ 32x3x3 $\rightarrow$ ReLU $\rightarrow$ $\text{MaxPool}_{(2,2)}$ 2x2 $\rightarrow$ Dropout(0.25) $\rightarrow$ $\text{Conv}_{(2,2)}$ 64x3x3 $\rightarrow$ ReLU $\rightarrow$ $\text{Conv}_{(2,2)}$ 64x3x3 $\rightarrow$ ReLU $\rightarrow$ $\text{MaxPool}_{(2,2)}$ 2x2 $\rightarrow$ Dropout(0.25) $\rightarrow$ FC 1024 $\rightarrow$ ReLU $\rightarrow$ FC 10

In the CIFAR dataset was randomly split between 50 clients, and each round 10 clients participated for a total of 3500 FL training rounds. We use 20% of the original training data as the validation set. We used SGD as the optimiser with a learning rate of 0.01 and batch size of 32. Each FL round a participating client locally trains for one epoch before dispatching their updated weights.

### A.3   Merged-FEMNIST

For the Merged-MNIST dataset we used the following neural network from the work in [19]:

$\text{Conv}_{(1,1)}$ 32x5x5 $\rightarrow$ ReLU $\rightarrow$ $\text{MaxPool}_{(2,2)}$ 2x2 $\rightarrow$ $\text{Conv}_{(2,2)}$ 64x5x5 $\rightarrow$ ReLU $\rightarrow$ $\text{MaxPool}_{(2,2)}$ 2x2 $\rightarrow$ FC 2048 $\rightarrow$ ReLU $\rightarrow$ FC 10

The Merged-FEMNIST is formed by merging classes corresponding to certain upper and lower case letters as described in [18] and is the data is divided into clients following the procedure in [19]. Each round 5 clients participated for a total of 3000 FL training rounds. We use 20% of the original training data as the validation set. We used SGD as the optimiser with a learning rate of 0.01 and batch size of 10. Each FL round a participating client locally trains for one epoch before dispatching their updated weights.

# Appendix B    Secret Share Times

Here we show supplementary results illustrating the time required by a client to compute the secret based on 1) the total number of clients the secret is split over and 2) the number of malicious clients that are present in the system.

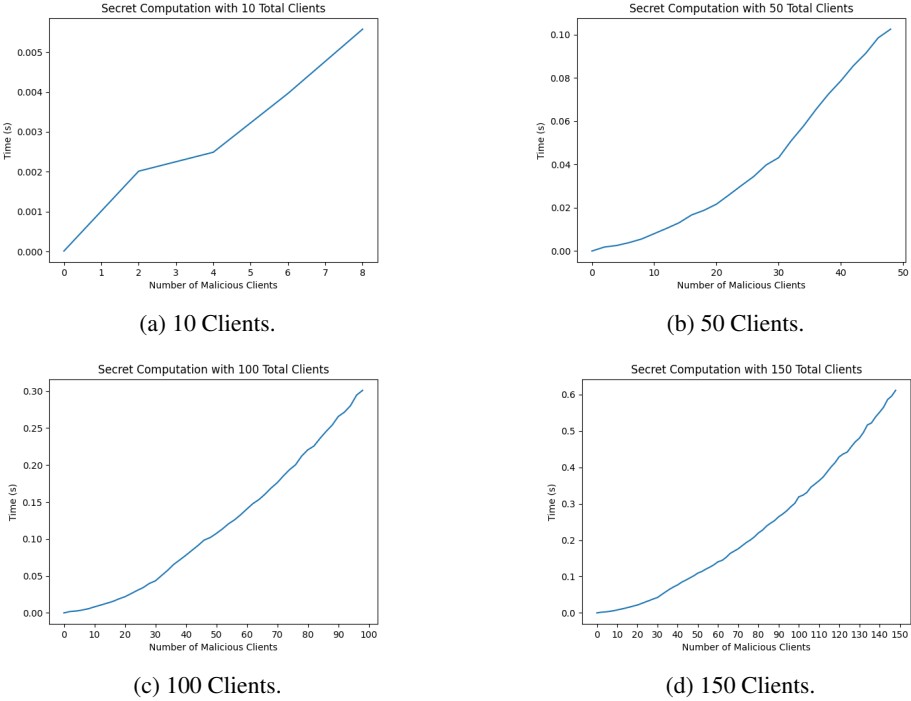

(a) 10 Clients.

(b) 50 Clients.

(c) 100 Clients.

(d) 150 Clients.

Figure 3: Shamir secret shares computation time based on the number of clients $n$ and the number of malicious clients $k$ considered.

