# OpenReview forum: "Verifiable Federated Learning"
_NeurIPS.cc/2022/Workshop/Federated_Learning — FL-NeurIPS 2022 Oral_

### Official Review · Reviewer_YmF1 · 2022-10-18
**Review of the paper**

This paper is a solid work that studies the problem of verifiable federated learning. The work discusses the setting that when clients are not the only party that can behave maliciously, but the aggregator itself may tamper the model. They develop a protocol which using binding commitments prevents an aggregator from modifying the resulting model, and only permits the aggregator to sum the supplied weights. They show through experiments that for large neural networks with millions of parameters the commitments can be computed in under 20 seconds.

Strengths:
- the problem studied in this paper is both interesting and important.
- The problem is quite novel and not well studied in the previous work, so it's great to present this and inform the researchers about it
- The paper is well written, easy for follow and understand
- The paper is technically sound (though, I recommend the authors to include the proofs in appendix for correctness)
- The presentation of the paper is well

Points to improve:
- Related work is well discovered and explained. It would be good and quite necessary to mention why this work is better than *all* of the prior work (only some selective related work are compared against, not all, so it makes the reader wonder maybe some prior work already addresses this problem similarly or even better)
- The experiments can be done on more realistic FL datasets (such as LEAF). Selecting cifar and mnist is okay, but for FL we already have several standard benchmarks. Also, in addition to computer vision datasets, also considering other ML problems is necessary.
- More explanation about the experiments is needed, e.g. the total number of parameters for each network, choice of networks, etc.
- Justify why AND if Intel i7-8700 CPU is representative of IoT or client devices? To me it seems like this CPU might be a pretty powerful CPU and perhaps not all clients are equipped with such CPUs
- The sentence "in the case of product recommendation, a service provider can have higher profits on certain items, and hence bias a aggregated model to recommend higher profit items, rather than those which would be the optimal recommendation for a client" does not carry a strong logic. A service provider theoretically is interested in recommending higher profit, and higher profit *is* the *optimal recommendation*.

---

### Official Review · Reviewer_NvzB · 2022-10-18
**good paper for defending the macilious server**

The paper considers a scenario where the users could verify whether the server has been honestly followed the aggregation protocol in federated learning.  The authors propose a lightweight verification protocol to avoid the server tampering with model to fulfill an attack objective.

The paper is well motivated. There are recent work confusing on the malicious server case in FL and showing that even under secure aggregation protocol, if there is a malicious server, it could still infer information on individual private training datasets by tampering the weights. This paper proposes a method from the cryptographic side, to avoid such vulnerability in the FL learning process.

As I am not an expert from the crypto, I could not really verify the correctness of the protocol. But the results seem to be solid. I think the protocol is allowed as well the usage of secure aggregation protocol, which avoids the server to infer private information directly from the received weights. The authors could add some discussion on the secure aggregation protocol in the related work.

Just few questions: for the time considered in Figure 2, is the time of executing function commit (eq 3) in a i7 CPU? The time might be longer when performs the function in a mobile device? Might be interesting to see such a statistic.

---

### Official Review · Reviewer_u1jR · 2022-10-18
**Paper 19 review**

This paper suggest a protocol which permits to only perform summation of the aggregated clients’ updates in federated learning. Thus it prevents the server to modify the model in a malicious way.

I am not an computer security expert, which seems to be the main subject of this work. Thus I have comments mainly from an outsiders view.

Firstly, I would like to ask the authors to discuss the motivation behind their work problem in more details. Based on the current introduction text it is not fully obvious how relevant is this issue for current and future federated learning systems.

Secondly, I think that the protocol’s description is not very clear for a person not very familiar with cryptography. Thus making it more accessible would be beneficial for a larger federated learning audience.

Lastly, a discussion on how the proposed technique is compatible with other standard methods like Gaussian mechanism (differential privacy), secure aggregation, compression (quantization, sparsification), adaptive optimization (like FedAdam optimizer, which not only averages but also performs a gradient-type step on the server) often essential for federated learning setting.

Overall, this submission creates an impression of a solid work which can tackle an important problem, but it lacks explanations and connections to contemporary federated learning literature.

---

### Decision · Program_Chairs · 2022-10-20

Accept (Oral)